# Stall force measurement of the kinesin-3 motor KIF1A using a programmable DNA origami nanospring

**Nobumichi Takamatsu[1,2], Hiroko Furumoto[3], Takayuki Ariga[4], Mitsuhiro Iwaki[4,5,6]\*, Kumiko Hayashi[1,2]\***

[1]The Institute for Solid State Physics, The University of Tokyo, Chiba, Japan; [2]Department of Complexity Science and Engineering, Graduate School of Frontier Sciences, The University of Tokyo, Chiba, Japan; [3]Systems Biochemistry in Pathology and Regeneration, Yamaguchi University Graduate School of Medicine, Ube, Japan; [4]Graduate School of Frontier Bioscience, The University of Osaka, Osaka, Japan; [5]Advanced ICT Research Institute, National Institute of Information and Communications Technology, Kobe, Japan; [6]Immunology Frontier Research Center (IFReC), The University of Osaka, Osaka, Japan

**\*For correspondence:**
iwakim@nict.go.jp (MI);
hayashi@issp.u-tokyo.ac.jp (KH)

**Competing interest:** The authors declare that no competing interests exist.

## eLife Assessment

Optical tweezers have been instrumental to the determination of mechanical parameters of molecular motors. This study by Takamatsu et al. reports key mechanical parameters of kinesin KIF1A using fluorescence microscopy, wherein the motor is tethered to a DNA nanospring, without the use of an optical trapping apparatus, which represents an exciting development. The approach and the findings reported change current thinking about KIF1A-mediated transport, with potential implications for understanding human disease. The findings are **important** and the strength of the evidence is **compelling**.

**Abstract** DNA origami technology is a method for designing and constructing nanoscale structures using DNA, and it is being applied across various fields. This technology was advanced by developing the nanospring (NS), a fluorescently visible molecular spring that quantifies forces through its extension and has been used to measure myosin-generated forces. This study aims to measure the force exerted by the kinesin-3 motor protein KIF1A, mutations of which cause KIF1A-associated neurological disorder (KAND) and are associated with reduced force and motility. Unlike kinesin-1, KIF1A detaches easily under perpendicular loads, which can occur in optical tweezers experiments. By applying force parallel to the microtubule using the NS, we were able to precisely measure the stall force even for KAND mutants, for which such measurements are typically challenging. This result highlights the potential of the NS as a new tool for force spectroscopy in biophysics.

## Introduction

The kinesin superfamily of microtubule-based motor proteins in humans consists of approximately 45 types and represents a large group of motor proteins with diverse functions (*Hirokawa et al., 2010*; *Vale, 2003*). Among them, KIF1A, a member of the kinesin-3 subfamily, plays a critical role in the long-distance transport of synaptic vesicle precursors and other cargos in neurons (*Gabrych et al.,*

*2019*). This transport is essential for maintaining synaptic activity, supporting neuronal development, and ensuring proper neuronal function. Structurally, KIF1A contains a motor domain that binds to microtubules and hydrolyzes ATP to generate mechanical force, driving movement toward the plus end of microtubules. In recent years, KIF1A has gained significant attention due to the identification of over 100 mutations linked to KIF1A-associated neurological disorder (KAND), a spectrum of neuro-developmental conditions characterized by spastic paraplegia, intellectual disability, and progressive neurodegeneration (*Kaur et al., 2020*; *Boyle et al., 2021*).

Single-molecule measurements of KIF1A mutants associated with KAND have developed (*Budaitis et al., 2021*; *Lam et al., 2021*; *Anazawa et al., 2022*; *Chiba et al., 2019*). Precise measurements of the velocity of KIF1A single molecules have been achieved using TIRF microscopy, while optical tweezers have enabled force measurements. These techniques have been used to compare the biophysical properties of KIF1A single molecules with those of its mutants that cause KAND. The motility of the single molecules has been reported to correlate with disease severity, highlighting the importance of single-molecule biophysical measurements into precision medicine (*Rao et al., 2025*). The velocity of KIF1A has been extensively studied; however, there is still room for improvement in force measurements using optical tweezers. To date, optical tweezers have been used to measure the stall force, defined as the maximum force that a motor can generate when its movement is halted, of various microtubule-based motor proteins (*Schnitzer et al., 2000*; *Gennerich et al., 2007*; *Elshenawy et al., 2019*; *Mallik et al., 2004*; *Tomishige et al., 2002*; *Okada et al., 2003*; *Nishiyama et al., 2002*). Since a submicron-sized bead is attached to the kinesin (~10 nm) and force is applied via the bead to the kinesin in this method, the applied force tends to pull it upward, which can sometimes lead to its detachment from the microtubule (*Pyrpassopoulos et al., 2020*; *Pyrpassopoulos et al., 2023*). Since KIF1A is easier to detach from microtubules compared to conventional kinesin-1, it is more challenging to successfully stall KIF1A in optical tweezer experiments without detachment. Indeed, most of the stalls lasted less than 1 s, and the force trajectories exhibited sawtooth-like patterns in previous studies (*Budaitis et al., 2021*; *Lam et al., 2021*).

To overcome the challenges associated with force measurements in KAND mutants, we employed an alternative approach based on DNA nanotechnology. In a previous study, Iwaki et al. developed a distinct method for force measurement by using programmable DNA origami to construct nanoscale, coil-like springs known as nanosprings (NSs) (*Iwaki et al., 2016*; *Matsubara et al., 2023*). These NSs are fluorescently labeled, allowing their extension to be visualized with conventional fluorescence microscopy. Since the force–extension relationship of the NS is pre-calibrated using acoustic force spectroscopy (AFM; *Matsubara et al., 2023*), the force exerted on the spring can be quantitatively estimated from its measured extension. In this study, we applied the NS method to measure the force generated by single KIF1A molecules, including both wild-type and KAND mutants. Using the NS system, we were able to apply a horizontal load parallel to the microtubule axis to single KIF1A molecules and clearly observed sustained stalling for several tens of seconds with reduced detachment. Notably, stalling events were also stably observed in KAND mutants such as P305L and V8M, which disrupt critical force-generating regions of the protein (e.g., the K-loop and neck linker) (*Budaitis et al., 2021*; *Lam et al., 2021*). We also included the A255V mutant, previously unmeasured in force assays, which is known to affect ATP hydrolysis (*Budaitis et al., 2021*). Furthermore, the NS allowed for precise stall force measurement, revealing force differences between KAND mutant homodimers and heterodimers composed of a wild-type and the mutant subunit. These results highlight the advantage of our approach in minimizing the force component perpendicular to the microtubule's long axis.

In previous single-molecule experiments in kinesin, DNA origami was used to control the number of cooperatively moving kinesin motors, and their velocities have been analyzed (*Furuta et al., 2013*; *Derr et al., 2012*). As a distinct application of DNA-origami-based motor protein systems, in this study, we successfully measured force using only fluorescence imaging. This study is significant in that it extends fluorescence imaging from merely capturing velocity to also allowing the quantification of force. In the field of motor proteins research, measurements of physical quantities related to distance, such as velocity and stepping motion, have rapidly advanced with the development of cutting-edge microscopy techniques like a super-resolution microscopy technology MINFLUX (*Balzarotti et al., 2017*), enabling their investigation not only in glass chambers but also inside living cells (*Deguchi et al., 2023*). However, since both force and velocity are crucial physical parameters to understand the functions of motor proteins, and force–velocity relationships are indeed related to their ATP hydrolysis

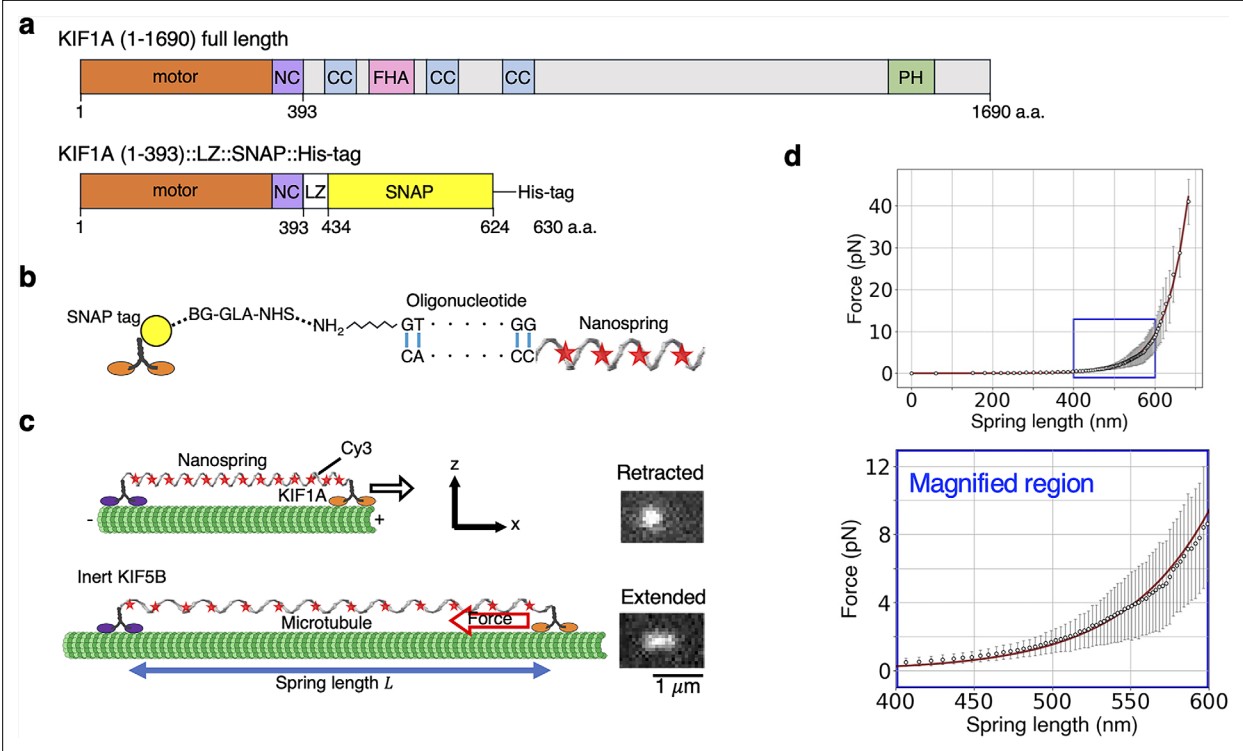

**Figure 1.** Experimental design for the stall force measurement of KIF1A using an NS. (**a**) Schematic of the domain structure of full-length KIF1A and the recombinant construct used in the experiments. To stabilize KIF1A dimers, which do not form stably without cargo-binding domains, a leucine zipper was incorporated, and SNAP-tags were added at the N-termini to enable chemical coupling to the NS. (**b**) Schematic of the chemical modifications required for coupling the NS to kinesin (Methods). (**c**) An inert KIF5B is anchored to the microtubule, and the NS extends as a KIF1A moves toward the plus end. The NS is uniformly labeled with Cy3 fluorophores, allowing force to be calculated from its extension. The micrographs depict the NS in the retracted and extended states. Here, the microtubule axis is defined as the $x$-direction, and the direction perpendicular to the microtubule is defined as the $z$-direction. (**d**) Force–extension relationship of the NS, showing nonlinear elastic behavior, calibrated by acoustic force spectroscopy (AFS) (*Matsubara et al., 2023*) and fitted with an exponential function (Methods).

The online version of this article includes the following source data and figure supplement(s) for figure 1:

**Source data 1.** Excel file containing force–extension measurements of the nanospring.

**Figure supplement 1.** Purification of recombinant SNAP-tagged KIF1A and KIF5B proteins.

**Figure supplement 2.** Purification of recombinant SNAP-tagged KIF1A heterodimers.

mechanisms (*Schnitzer et al., 2000*; *Sasaki et al., 2018*), we believe that the development of a new type of force measurement is of great significance in this field. The development of NS-based force measurement, which enables force quantification at the nanoscale, is expected to have significant implications not only in the field of motor protein research but also in the broader study of protein biophysics.

## Results

### The NS–KIF1A–inert KIF5B complex as a force measurement system

To measure the force generated by single KIF1A molecule, we constructed a complex comprising a KIF1A molecule, an inert KIF5B mutant, and an NS. For KIF1A, the motor domain (amino acid residues 1–393) was used, excluding the C-terminal and cargo-binding domains (*Figure 1a*). Since KIF1A does not form stable dimers in the absence of cargo-binding domains, a leucine zipper domain was incorporated to stabilize KIF1A dimers, mimicked the activated state of full-length KIF1A, as in the previous study (*Budaitis et al., 2021*). As for the inert KIF5B attached to the one end of the NS, it was engineered to remain immobilized on microtubules by introducing the G234A mutation (Methods). This mutant lacks ATP hydrolysis activity and, as a result, does not exhibit motility, allowing it to remain

stably attached to microtubules (*Rice et al., 1999*). Both kinesins were equipped with SNAP-tags at their N-termini, which enabled chemical coupling to the NS (*Figure 1b*; Methods). The inert KIF5B mutant served to anchor the KIF1A–NS complex to a microtubule. This anchoring of the KIF5B to the microtubule allowed a KIF1A to stay near the microtubule, increasing the probability of its binding to the microtubule.

At equilibrium, in the absence of tension applied by KIF1A, the NS, uniformly labeled with 124 Cy3 fluorophores, exhibited a resting length of approximately 300 nm and appeared as a diffraction-limited circular spot in a fluorescence image (*Figure 1c*). In the presence of ATP, the NS extended as the KIF1A moved against the load, and its fluorescence image became an elongated shape (*Figure 1c*). To calibrate the pulling force acting on the NS from its extension, the AFS was used as in the previous study (*Matsubara et al., 2023*). *Figure 1d* presents the relationship between NS extension and force measured by the AFS, revealing its nonlinear elastic behavior. Its nonlinear force–extension relationship was fitted with an exponential function (Methods).

## Simulation-based evaluation of fluorescence image analysis methods

Before analyzing the fluorescence images of an NS, we generated artificial images that mimic experimental conditions in order to develop a method for estimating the length of a rod-shaped object from its fluorescence image. The artificial images were generated by placing 116 two-dimensional isotropic Gaussian functions aligned along a straight line (*Figure 2a*), to mimic a DNA calibration rod labeled with 116 fluorescent molecules, as introduced in the following section. The value of the variance in the Gaussian function was determined based on the approximate shape of the point spread function (PSF) of the Cy3 fluorophore (*Figure 2b*). While the PSF is, in principle, influenced by the optical system, here it was used as a reference value for generating the artificial images. White noise was then added to the images. When the 116 Gaussian functions were superimposed, they appeared as a single bright spot with an elliptical shape (*Figure 2c*, top). For this artificial image, we present a comparison of the results obtained using two methods: the Gaussian fitting method and the chain fitting method.

In the Gaussian fitting method, the whole fluorescence spot is approximated using a single two-dimensional Gaussian function $I_g(x, y)$, and the estimated standard deviation along the long axis ($\sigma_{\text{long}}$) (*Figure 2c*, middle).

$$I_g(x, y) = C_g \exp\left[-\frac{(x - x_0)^2}{2\sigma_{\text{long}}} + \frac{(y - y_0)^2}{2\sigma_{\text{short}}}\right]. \tag{1}$$

Here, $C_g$ is a constant, and $x$ and $y$ represent the major and minor axis directions, respectively. *Figure 2d* (gray symbols) shows the relationship between the true length $L$ of the rod-shaped object and $\sigma_{\text{long}}$. If the linear relationship between $L$ and $\sigma_{\text{long}}$ is determined in advance, the value of $L$ can be estimated from the measured $\sigma_{\text{long}}$ (*Matsubara et al., 2023*).

While the Gaussian fitting method provided reasonable results, we considered an alternative approach, called the chain fitting method, to directly estimate the length $L$ of the model. In the Gaussian fitting method, proposed in the previous study (*Matsubara et al., 2023*), the true length $L$ of the rod is not included as a parameter in the fitting function (*Equation 1*). The chain fitting method is designed based on the structure in which fluorescent molecules are arranged in a line, and because the fitting parameters directly include the length $L$, it is expected to provide a more accurate estimation than the Gaussian fitting method. The fluorescence intensity of this model can be described by the following equation:

$$\begin{aligned} I_c(x, y) &= C_c \int_{-\frac{L}{2}}^{\frac{L}{2}} \exp\left(-\frac{(x - x')^2 + y^2}{2\sigma^2}\right) dx' \\ &= \sqrt{\frac{\pi}{2}} C_c \sigma \cdot \exp\left(-\frac{y^2}{2\sigma^2}\right)\left[\Phi\left(\frac{x + \frac{L}{2}}{\sqrt{2}\sigma}\right) - \Phi\left(\frac{x - \frac{L}{2}}{\sqrt{2}\sigma}\right)\right], \end{aligned} \tag{2}$$

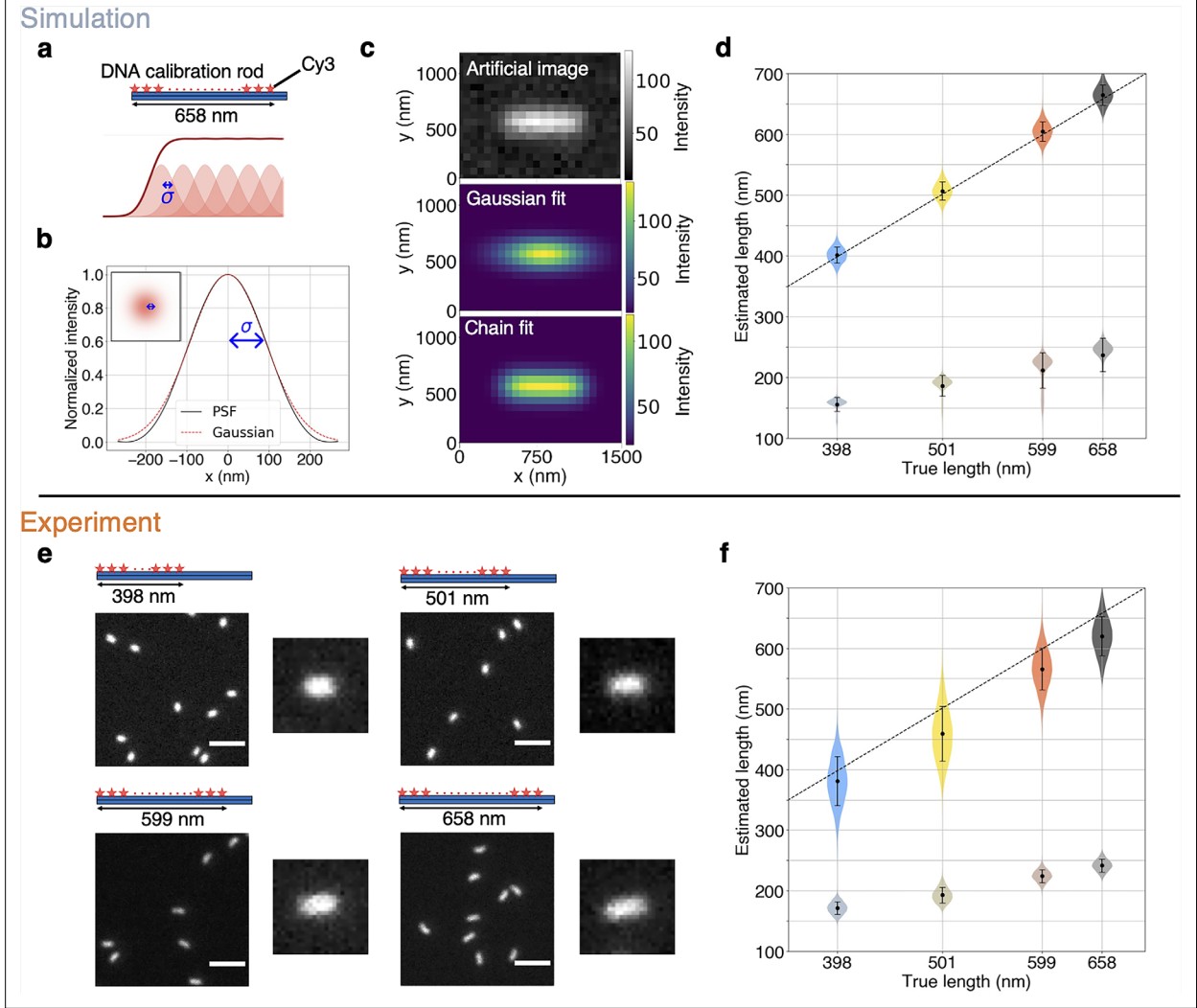

**Figure 2.** Length estimation using DNA calibration rods. (**a**) The fluorescence intensity of the DNA calibration rod was theoretically modeled as a superposition of Gaussian functions with variance $\sigma^2$ aligned along a straight line, and an artificial image was generated accordingly. (**b**) The $\sigma$ value of the Gaussian function used in the model was fitted to match the point spread function of Cy3 fluorescent dye (black line), resulting in a value of 90.65 nm. (**c**) Artificial image generated by placing 116 two-dimensional Gaussian functions (top). When 116 Gaussian functions were superimposed, they appeared as a single bright spot with an elliptical shape. The image (top) was fitted by the Gaussian fitting method (**Equation 1**) (middle) and the chain fitting method (**Equation 2**) (bottom). (**d**) Estimated length plotted against the true length of the model ($L$) using the Gaussian fitting method (**Equation 1**) (dark colors) and the chain fitting method (**Equation 2**) (bright colors), respectively. The dotted line represents the linear equation $y = x$. The chain fitting model provides estimates of the true value of $L$ in the simulation. Note that we generated 30 artificial videos, each consisting of 300 frames. (**e**) Fluorescence micrographs of the DNA calibration rods (**Matsubara et al., 2023**) obtained in real experiments, with lengths of 398, 501, 599, and 658 nm. The scale bars indicate $2~\mu\mathrm{m}$. (**f**) Estimated length plotted against the true length of the rods using the Gaussian fitting method (**Equation 1**) (dark colors) and the chain fitting method (**Equation 2**) (bright colors), respectively. The dotted line represents the linear equation $y = x$. The chain fitting method provides estimates closer to the true value of $L$. Note that approximately 30 videos were recorded, each containing about 300 frames for each DNA calibration rod.

The online version of this article includes the following source data for figure 2:

**Source data 1.** Excel file containing estimated lengths of fluorescence images of DNA rods for simulations and experiments.

where $C_c$ is a constant, and $\Phi$ is the error function. *Figure 2c*, bottom, shows the resulting image obtained by the chain fitting method. The simulation results indeed demonstrated that the estimated length by the chain fitting method corresponded with the true length $L$ (*Figure 2d*, color symbols).

## Experimental validation of the chain fitting method using DNA calibration rods

We experimentally validated our approach of the chain fitting method using DNA calibration rods with known lengths (*Matsubara et al., 2023*). *Figure 2e* shows the fluorescence images of biotinylated DNA calibration rods immobilized on a glass surface via biotin–BSA and streptavidin (Methods), recorded under the same conditions used for observation of the NSs. As illustrated in the schematic, a total of 116 Cy3 molecules were conjugated to the DNA calibration rod at equal intervals (*Matsubara et al., 2023*). The length was varied by adjusting the interval between fluorescent dyes. The end-to-end distances between fluorescent molecules were designed to be 398, 501, 599, and 658 nm, respectively. Since the NSs were labeled with 124 Cy3 fluorescent molecules, we considered the fluorescence images obtained from these rods to be similar to those of the NSs.

*Figure 2f* shows the $\sigma_{\text{long}}$ and length $L$ estimated from experimental fluorescence images of DNA calibration rods using the Gaussian fitting method (*Equation 1*) and chain fitting method (*Equation 2*), respectively. There is a proportional relationship between $\sigma_{\text{long}}$ and the true length. On the other hand, $L$ estimated by *Equation 2* provides a direct estimate close to the true length of the DNA calibration rod. However, the fact that $L$ was approximately 10% shorter than the true length is considered to be likely due to slight bending of the DNA-origami structures caused by thermal fluctuations. Indeed, in the case of longer rods shown in *Figure 2e*, a slight bending was observed. Accordingly, the length was corrected by a factor of 1.07 (Methods). In the following, the extension of the NS was primarily estimated using the chain fitting method. Note that results obtained using the conventional Gaussian fitting method are also presented in *Figure 3—figure supplement 2*. The results from both methods were consistent.

## Observation of stall behavior in wild-type KIF1A

*Video 1* shows a representative example of the movement of an NS with KIF1A and KIF5B. At first, an inert KIF5B is firmly fixed to the microtubule. Then, the diffusive behavior of a KIF1A—attached to the NS, which is anchored to the microtubule via the inert KIF5B—as it searches for a binding site on the microtubule, is observed. Once the KIF1A binds to the microtubule, it starts moving along the microtubule and stretches the spring. When the tension in the NS balances with the maximum force generated by the KIF1A, the extension temporarily halts during the time interval (attachment duration). Eventually, the KIF1A detaches from the microtubule, the tension is released, and the NS returns to its retracted state. This cycle event was repeatedly observed.

The extension of the NS was estimated by the chain fitting method (*Equation 2*; *Figure 3a*, bottom). For the event during a time interval ($2.8\,\text{s} \leq t \leq 21.2$ s), the histogram of the spring length was calculated (*Figure 3a*, left). The histogram exhibits a bimodal Gaussian distribution, with each peak corresponding to the extended and retracted states of the NS shown in the micrographs (*Figure 3a* left). $L_{\text{stall}}$ was calculated by averaging the NS extension during the force-plateau periods within each attachment event. The attachment of KIF1A to the microtubule was identified by the marked suppression of angular fluctuations of the NS relative to the microtubule (Methods and *Figure 3a*, top), which clearly distinguished the bound state from the diffusive search behavior. Within this attachment duration, the force plateau (stall duration $\Delta t$) was detected by analyzing the rate of relative increase in NS's length (Methods). The mean NS extension over these plateau segments was defined as $L_{\text{stall}}$.

In our experiments, we analyzed a total of 86 events from 24 KIF1A molecules (*Figure 3b*, left). From the mean value of $L_{\text{stall}}$ ($562 \pm 48\,\text{nm}$), the stall force of the wild-type KIF1A was estimated to be 4.7 pN (*Table 1*) by using the force–extension calibration of the NS (*Figure 1d*). This value is similar to the stall force values of the wild-type KIF1A(1–393) measured previously (*Pyrpassopoulos et al., 2023*), and larger than the detachment force values (*Budaitis et al., 2021*; *Lam et al., 2021*). The right panel of *Figure 3b* shows the

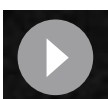

**Video 1.** Wild-type KIF1A pulling a nanospring. The video is shown at 10× speed.
https://elifesciences.org/articles/108477/figures#video1

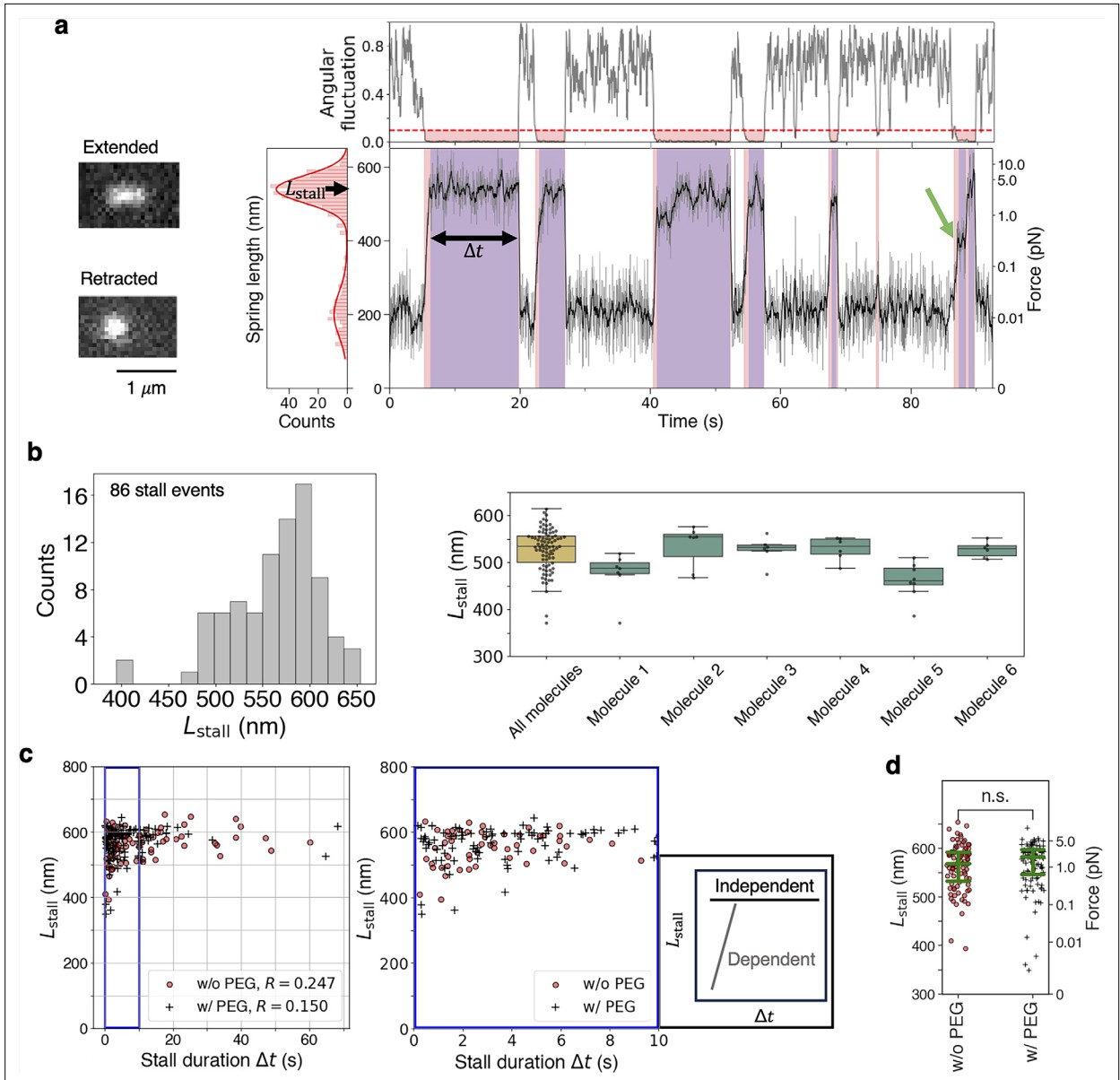

**Figure 3.** Stall force measurement of wild-type KIF1A homodimers using NSs. (**a**) Time course of NS extension ($L(t)$) in the case of wild-type KIF1A. The black line (trace) represents the average over 10 frames. As illustrated in the schematic in **Figure 1c**, the NS is stretched (micrograph, top) as a KIF1A moves toward the plus end of the microtubule. When the load reaches the maximum force that the KIF1A can generate, a stall is observed, followed by the detachment of the KIF1A from the microtubule. The NS then returns to its original retracted state (micrograph, bottom). The stall duration $\Delta t$ (violet region) was defined based on the angular fluctuation and the rate of relative increase in NS's length (Methods), where the red regions represent the attachment durations decided based on the angular fluctuations. For each stall event, the histogram of NS extension exhibits a bimodal Gaussian distribution, with the higher peak corresponding to the stall length $L_{\text{stall}}$ (left panel). (**b**) Histogram of $L_{\text{stall}}$ values calculated from 86 stall events (left). The right panel shows the distribution of stall forces for each KIF1A molecule in which six or more stall events were observed. (**c**) $L_{\text{stall}}$ as a function of stall duration $\Delta t$ with ($n = 105$) and without ($n = 86$) PEG. The correlation coefficient ($R$) between $L_{\text{stall}}$ and $\Delta t$ is shown in the figure (left). The right panel presents the magnified view of the blue rectangle in the left panel and clearly indicates that $R$ is small. (**d**) Comparison of $L_{\text{stall}}$ and $\Delta t$ with and without PEG (Mann–Whitney $U$ test, p = 0.4718 for $L_{\text{stall}}$, p = 0.0616 for $\Delta t$). n.s., not significant (p ≥ 0.05). The green bars indicate the median values along with the first and third quartiles.

The online version of this article includes the following source data, source code, and figure supplement(s) for figure 3:

**Source code 1.** Time-course analysis of single-molecule videos.

**Source data 1.** Excel file containing the time course, $L_{\text{stall}}$ and $\Delta t$ from the stall force experiment of wild-type KIF1A.

**Figure supplement 1.** Time course of a nanospring (NS) extension at a recording rate of 100 fps for wild-type KIF1A homodimers.

*Figure 3 continued on next page*

*Figure 3 continued*

**Figure supplement 2.** Estimation of NS extensions using the Gaussian fitting method (*Equation 1*, main text).

**Figure supplement 3.** Cumulative distributions of stall duration ($\Delta t$) in the case of the KIF1A(WT).

distribution of stall forces for each KIF1A molecule in which six or more stall events were observed. The broad distribution of stall forces arises not only from differences between molecules, but also from the variability of stall forces within a single molecule. The broad distribution is caused by the presence of step-back events (e.g., a green arrow in *Figure 3a*); however, the chemical state underlying these step backs in KIF1A remains unknown.

In the results presented here, the extension of the NS was estimated using the chain fitting method (*Equation 2*). For comparison, we also present the results obtained using the conventional Gaussian fitting method (*Equation 1*) in *Figure 3—figure supplement 2*. The two methods provided consistent values, suggesting that future analyses can be performed without the need for prior calibration using DNA rods, which is required by the Gaussian fitting method. In addition, while direct velocity measurements were difficult in the NS-based experiments, we analyzed the behavior under low-load conditions by overlaying the time courses of 10 stall events (*Figure 4—figure supplement 1*).

## Stall duration $\Delta t$

In previous studies (*Budaitis et al., 2021*; *Lam et al., 2021*), most of the stalls lasted less than 1 s, and the force trajectories exhibited sawtooth-like patterns, which was a characteristic feature of force generation by KIF1A unlike kinesin-1, which exhibits clear stalling behavior (*Kojima et al., 1997*). However, in our experiments, we observed clear stall states lasting for several tens of seconds, similar to the stalling behavior observed with kinesin-1 (*Figure 3a*). To quantify this observation, we measured the stall durations ($\Delta t$) and analyzed the relationship between $L_{stall}$ and $\Delta t$ (*Figure 3c*).

In a single-bead optical tweezers experiment, it was reported that $\Delta t$ was correlated with the detachment force when the detachment of a KIF1A from the microtubule was likely accelerated by forces applied perpendicular to the microtubule's long axis ($z$-axis depicted in the schematics of *Figure 1c*), due to the contact between a single bead and an underlying microtubule (*Pyrpassopoulos et al., 2020*). By using the three-bead optical tweezers system (*Pyrpassopoulos et al., 2020*;

**Table 1.** Stall force of KIF1A(1–393) estimated from the average value of $L_{stall}$.
The stall force values were calculated from the mean values of $L_{stall}$ by using the force–extension relationship of the NS (*Figure 1d*). The error of $L_{stall}$ value represents the standard deviation (SD).

|  | Stall force (pN) | Detachment force (pN) | Termination force (pN) |
|---|---|---|---|
| WT/WT | 4.7 ($n = 86$)<br>($L_{stall} = 562 \pm 48$ nm) | 2.7[*]<br>2.2[†] | 4 [‡]<br>6 [§] |
| P305L/P305L | 0.2 ($n = 58$)<br>($L_{stall} = 375 \pm 49$ nm) | 0.7[†] | - |
| P305L/WT | 0.3 ($n = 41$)<br>($L_{stall} = 402 \pm 54$ nm) | - | - |
| V8M/V8M | 2.2 ($n = 95$)<br>($L_{stall} = 516 \pm 57$ nm) | 1.9[*] | - |
| V8M/WT | 1.0 ($n = 61$)<br>($L_{stall} = 469 \pm 27$ nm) | - | - |
| A255V/A255V | 3.0 ($n = 83$)<br>($L_{stall} = 535 \pm 46$ nm) | - | - |
| A255V/WT | 2.5 ($n = 40$)<br>($L_{stall} = 524 \pm 26$ nm) | - | - |

Detachment force and termination force measured by using optical tweezers.
[*]Reported in reference (*Budaitis et al., 2021*).
[†]Reported in reference (*Lam et al., 2021*).
[‡]Reported in reference (*Pyrpassopoulos et al., 2023*) for the single-bead assay.
[§]Reported in reference (*Pyrpassopoulos et al., 2023*) for the three-bead assay.

*Pyrpassopoulos et al., 2023*), which was a modified version of the conventional optical trapping assay, to minimize the $z$-component of the force, it was found that $\Delta t$ did not depend on the detachment force (schematic in *Figure 3c*; *Pyrpassopoulos et al., 2020*). Our experimental results also indicated that $\Delta t$ was independent of $L_{\text{stall}}$, as the correlation coefficient was 0.2 (*Figure 3c*), suggesting that the $z$-component of the force was effectively minimized.

## Effect of electrostatic repulsion on force measurement

Because the DNA-based NS is negatively charged, we investigated the effect of its electrostatic repulsion with negatively charged microtubules. This repulsion could potentially promote the detachment of KIF1A from the microtubule, raising a concern for the measurement. To minimize repulsive interactions between the NS and the microtubule, we coated the NS with oligolysine conjugated to polyethylene glycol (PEG) to render the spring electrically neutral (Methods). Based on a statistical test, there was little difference in $L_{\text{stall}}$ and stall duration ($\Delta t$) between the conditions with and without PEG (*Figure 3c, d*; Mann–Whitney $U$ test, p = 0.4718 for $L_{\text{stall}}$, p = 0.0616 for $\Delta t$). These results suggest that the negative charge of the NS had little influence on the stall force measurements.

## Stall force measurement of the KIF1A disease-associated variant P305L

P305 of the motor domain of KIF1A is located adjacent to the positively charged K-loop insertion region of KIF1A (*Figure 4a*). The positively charged K-loop interacts with the negatively charged C-terminal loop of tubulin in microtubules, promoting a high binding affinity to microtubules (*Tomishige et al., 2002*; *Soppina and Verhey, 2014*). Therefore, a mutation at P305 directly affects the interaction between KIF1A and microtubules. In other words, the P305L mutation could impair the function of the K-loop, leading to a reduced binding affinity to microtubules (*Lam et al., 2021*; *Anazawa et al., 2022*). It was reported that the velocity and run length were reduced by half, and the force decreased fourfold, indicating that the P305L mutation impaired the motility of the motor (*Lam et al., 2021*). The decrease in force generation in the P305L mutant is a serious defect.

The wild-type KIF1A depicted in *Figure 1b, c* was replaced with the P305L mutant, and force measurements were performed using an NS for the mutant (*Figure 4b*), as shown in *Video 2*. Although it has been reported that the landing rate of P305L on microtubules was reduced by 97% (*Lam et al., 2021*; *Anazawa et al., 2022*), in this experiment, an inert KIF5B firmly anchored to the microtubule at the end of the NS, which kept the P305L mutant in close proximity to the microtubule, thereby effectively increasing its binding probability. In other words, the anchoring effect enabled clear observation of microtubule binding by the P305L mutant. This high binding rate is one of the advantages of the NS–KIF1A–inert KIF5B complex, when used as a force measurement system. Based on the value of $L_{\text{stall}}$, the force generated by the mutant was reduced to be 0.2 pN (*Table 1*). Note that the histogram of $L_{\text{stall}}$ is shown in *Figure 4—figure supplement 2*.

KIF1A is considered to primarily form dimers to anterogradely transport synaptic cargos in neurons. In recent years, the motility of heterodimers composed of wild-type and KAND-associated mutant subunits has begun to be investigated experimentally (*Budaitis et al., 2021*; *Anazawa et al., 2022*; *Kita et al., 2023*). In heterozygous patients, WT/WT, P305L/WT, and P305L/P305L dimers coexist, highlighting the importance of studying the heterodimers. As in the previous study (*Anazawa et al., 2022*), we constructed heterodimers composed of wild-type and P305L subunits (Methods) and performed stall force experiments using these heterodimers (*Figure 4c*, *Table 1*).

## Stall force measurement of the KIF1A disease-associated variants V8M and A255V

Like kinesin-1, KIF1A also utilizes conformational changes in the neck linker to generate force (*Budaitis et al., 2021*). The V8M mutation may hinder the neck linker from accessing its docking pocket, which in turn disrupts docking and impairs force generation (*Figure 4a*). Using the NS-based stall force measurement, we also observed that the stall force of the V8M homodimer was reduced by approximately half (*Figure 4d*, *Table 1*), consistent with the previous study (*Budaitis et al., 2021*).

The A255 residue is located in loop L11 of the switch II cluster and may affect KIF1A function by altering the structure of L11 within switch II or the back door (*Lee et al., 2015*; *Figure 4a*). In the single-molecule experiments, although its force has not yet been measured, a decrease in velocity has been reported (*Chiba et al., 2019*; *Guedes-Dias et al., 2019*). Loss of function in the switch II region

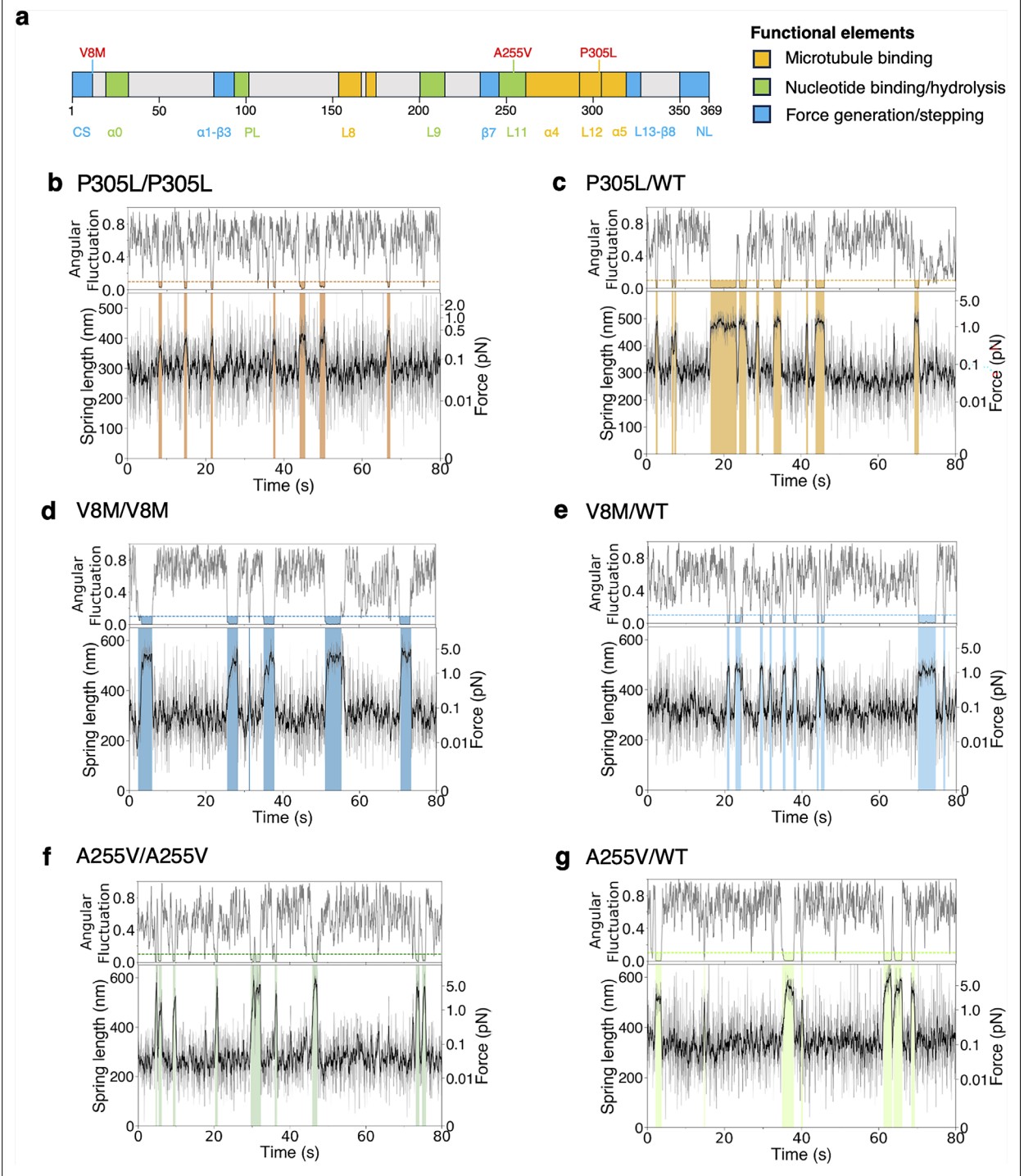

**Figure 4.** Stall force measurement of KAND mutants KIF1A homodimers and heterodimers using NSs. (**a**) Schematic of KIF1A domain structure showing the functions affected by the P305L, V8M, and A255V mutations (**Budaitis et al., 2021**). Representative traces of NS extension are shown for homodimers and WT-mutant heterodimers of P305L, V8M, and A255V (**b–g**). The black lines (traces) represent the average over 10 frames. The lighter-colored regions in the graph represent the attachment durations, while the darker-colored regions indicate the stall durations. The identification of these durations is described in the Methods section.

The online version of this article includes the following source data and figure supplement(s) for figure 4:

**Source data 1.** Excel file containing the time course data from the stall force experiment of KIF1A mutants.

**Figure supplement 1.** Displacement (calculated from the extensions of NSs) for KIF1A KAND mutants.

**Figure supplement 2.** Histograms of $L_{stall}$ for KIF1A KAND mutants.

**Video 2.** The P305L mutant pulling a nanospring. The video is shown at 1×speed.

https://elifesciences.org/articles/108477/figures#video2

or the back door may impair the conversion of ATP hydrolysis-driven conformational changes into mechanical force. Indeed, our NS-based stall force measurements revealed that the stall force was reduced by approximately half compared to the wild-type (*Figure 4f*, *Table 1*). We also performed stall force measurements for heterodimers composed of the KAND mutants V8M (*Figure 4e*, *Table 1*) or A255V (*Figure 4g*, *Table 1*) and the wild-type subunit. Note that the histogram of $L_{stall}$ for the mutants in this section is shown in *Figure 4—figure supplement 2*.

### $L_{stall}$ values for different KIF1A variants

All $L_{stall}$ values for the KIF1A mutants are summarized in *Figure 5a*. The values increased in the order of P305L, V8M, and A255V. We then converted the $L_{stall}$ values into forces and compared them with the clinical severity of KAND (*Figure 5—figure supplement 1*). Specifically, we compared them with REVEL, CADD v1.4, and the evolutionary scale model (ESM) score reported in the reference (*Boyle et al., 2021*). Among these metrics, the ensemble prediction score REVEL, which integrates multiple pathogenicity prediction algorithms, showed the correlation. It is important to expand the number of variants analyzed to further investigate the relationship between stall force and the severity of KAND.

### $\Delta t$ values for different KIF1A variants

As shown in *Figure 5b*, stall duration ($\Delta t$) was compared among KAND mutants (P305L, V8M, and A255V). Because stall behavior has rarely been observed for KIF1A in single-bead optical tweezers experiments, comparison of $\Delta t$ has been limited in previous studies (*Budaitis et al., 2021*; *Lam et al., 2021*). $\Delta t$ did not show as clear differences among the mutants as $L_{stall}$ did. For WT, which provided a sufficient number of stall events, we investigated the cumulative distribution of $\Delta t$ (*Figure 3—figure supplement 3*). Understanding KIF1A's behavior associated with these time constants will be important in future studies.

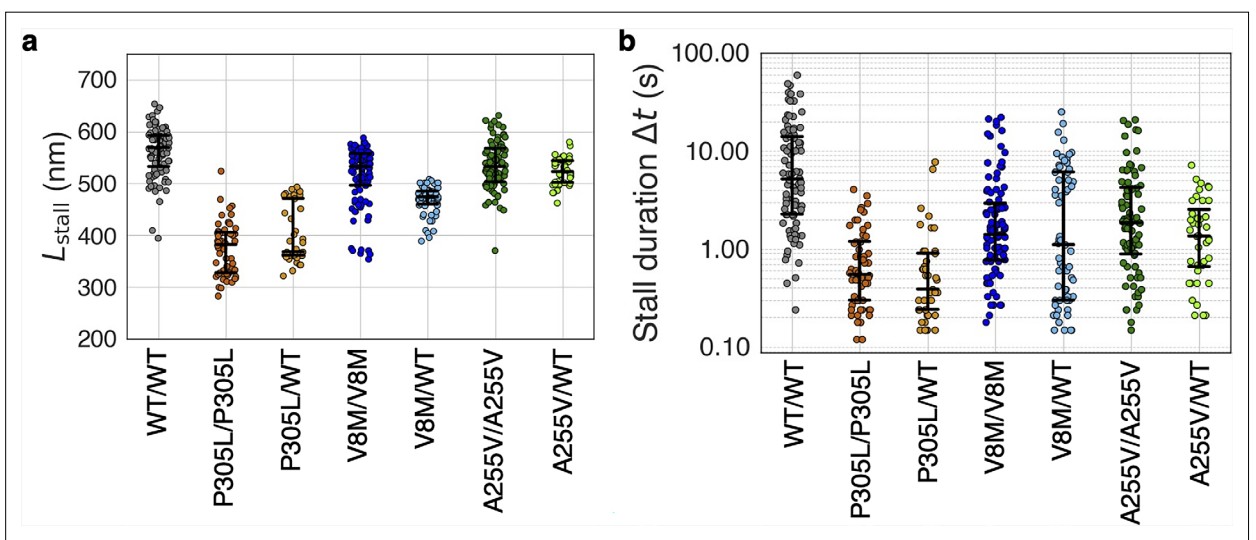

**Figure 5.** Comparison of $L_{stall}$ and $\Delta t$ among KAND mutants. $L_{stall}$ (**a**) and $\Delta t$ (**b**) for homodimers and heterodimers of P305L, V8M, and A255V, compared with WT.

The online version of this article includes the following source data and figure supplement(s) for figure 5:

**Source data 1.** Excel file containing $L_{stall}$ and $\Delta t$ from the stall force experiment of KIF1A mutants.

**Figure supplement 1.** Comparison between stall force and KAND severity scores.

## Discussion

We measured the force generated by KIF1A using the NSs developed by *Iwaki et al., 2016*; *Matsubara et al., 2023*; *Fujita et al., 2019*; *Figure 1*. Because KIF1A more readily detaches from microtubules compared to kinesin-1, and the $z$-component (refer to *Figure 1c* for the $z$-direction) of the applied forces often causes detachment in optical tweezers experiments, making it difficult to observe its stall behavior. In contrast, by using the NS, we were able to apply load to a single KIF1A molecule solely in the horizontal direction ($x$-axis) along microtubules, enabling clear observation of the stall events (*Figure 3*). Note that, while performing these measurements, we also improved the image analysis method (*Figure 2*). In a previous study (*Matsubara et al., 2023*), the length of the NS was estimated using the Gaussian fitting method (*Equation 1*). In contrast, in the present study, we adopted the chain fitting method (*Equation 2*), which allowed for a more direct estimation of the length $L$ from the images. Below, we discuss the advantages and future prospects of NS-based force measurements.

In KAND mutants, ESM scores have been compared with clinical phenotype scores such as the Vineland Adaptive Behavior Composite (VABS-ABC) scores (*Sudnawa et al., 2024*). ESM refers to large-scale protein language models based on Transformer architectures, trained on millions of sequences to predict structure, function, and mutation effects from primary amino acid sequences (*Rives et al., 2021*). However, due to the currently low correlation between ESM scores and clinical severity, incorporating biophysical parameters such as velocity and force may provide additional predictive insight, particularly for de novo mutations—a direction already explored in part by a previous study (*Rao et al., 2025*). In the present study, we propose a high-throughput method for accurately measuring the force generated by KIF1A mutants using the NSs. Anchoring a KIF1A–NS complex to the microtubule via an inert KIF5B facilitated the observation of stall events even for KAND-related KIF1A mutants (*Figure 4*), which were otherwise prone to detachment due to the dysfunctions caused by the mutations. Indeed, we successfully measured the stall force of several KAND-associated mutants, including P305L, V8M, and A255V. P305L, V8M, and A255V are representative mutants that primarily impair microtubule binding, force generation near the neck linker, and ATP hydrolysis near the Switch II region, respectively (*Figure 4a*). Notably, stall force measurements for the homodimeric A255V and heterodimeric combinations P305L/WT, V8M/WT, and A255V/WT had not previously been reported using optical tweezers. Building a database of such biophysical measurements will be essential for linking molecular dysfunction to clinical outcomes. The NS-based force assays offer a powerful platform for this purpose. In *Figure 5—figure supplement 1*, we additionally compared these three mutants with the REVEL, CADD v1.4, and ESM scores (*Boyle et al., 2021*). It is important to further investigate the relationship between stall force and the severity of KAND.

As noted in the introduction, optical tweezers have been the primary method for measuring forces generated by motor proteins (*Schnitzer et al., 2000*; *Gennerich et al., 2007*; *Elshenawy et al., 2019*; *Mallik et al., 2004*; *Tomishige et al., 2002*; *Svoboda et al., 1993*; *Finer et al., 1994*; *Kaya and Higuchi, 2010*; *Ariga et al., 2018*). An advantage of the NS is that it enables precise investigation of the force exerted by a single motor protein, whereas multiple kinesin molecules may attach to a bead in optical tweezers experiments. Moreover, unlike optical tweezers, which require individual manipulation of each bead and can measure the force of only one motor protein at a time, the NS-based assay allows simultaneous recording and force measurements of multiple NS–KIF1A complexes within the field of view. This results in significantly higher throughput and represents a key advantage over optical tweezers. From the perspective of compatibility with other measurement techniques, the near-infrared laser used in optical tweezers readily causes photobleaching of organic dyes, making single-molecule imaging of dye-labeled proteins challenging. For example, this means that optical tweezers cannot be used simultaneously with MINFLUX (*Deguchi et al., 2023*). Moreover, the NS can also be combined with high-speed AFM or electron microscopy, enabling structural analysis of proteins under quantitatively controlled mechanical load. In fact, although not NSs, DNA origami-based thick filaments with attached myosin have been investigated using AFM (*Fujita et al., 2019*). Therefore, the DNA origami-based NSs are also expected to be used with AFM.

Although NSs offer several advantages, they also have disadvantages. While the optical trap can be approximated as a linear spring, the NS behaves as a nonlinear spring (*Figure 1d*). Consequently, fluctuations in its length can sometimes lead to large errors in force measurements. Therefore, it is crucial to adjust the spring stiffness to match the force range generated by the motor protein. In this study, we used NSs with different physical properties from those used in the previous force

measurements on myosin VI (*Schnitzer et al., 2000*). Wild-type KIF1A generates a larger stall force than myosin, requiring a different dynamic range for the nonlinear behavior of the spring. Owing to DNA origami technology, the mechanical properties of the NS can be tuned to match the appropriate force range for measuring molecular stiffness.

The reason why the stall force exhibits a wide range of values may be not only the non-linearity of the NS, but also the complexity of the underlying kinetics of a single KIF1A, such as step-back events (e.g., the green arrow in *Figure 3a*). Indeed, analysis of the $\Delta t$ distribution (*Figure 3—figure supplement 3*) suggests that the motion during stall events can be characterized by two (or more) reaction rate constants (*Figure 3—figure supplement 3*). Determining the detailed chemical reaction model underlying these stall events will be an important subject for future studies.

With the recent development of the super-resolution microscope MINFLUX, live-cell tracking of single fluorescent molecules has become possible, enabling the precise observation of kinesin-1 stepping motion in living cells with nanometer spatial and millisecond temporal resolution (*Deguchi et al., 2023*). To date, models of chemo-mechanical energy conversion in motor proteins have been proposed based on the load dependence of their stepping motion (*Sasaki et al., 2018*). The intracellular environment is highly crowded with biomolecules and also contains active fluctuations (*Nishizawa et al., 2017*), resulting in complex viscoelastic properties and an inhomogeneous refractive index. This leads to challenges in using optical tweezers, such as accurately estimating the trap stiffness of laser-trapped vesicles in cells. On the other hand, the NSs offer a method of measuring force solely through fluorescence imaging, providing a potential solution to this problem. In relation to cellular applications, the NSs have already been used as force sensors to visualize the dynamics of integrins (*Matsubara et al., 2023*). Expanding such applications, the introduction of the NS–kinesin complex into living cells offers a promising approach to uncover the load-dependent dynamics of kinesin in its native environment.

# Materials and methods

## Key resources table

| Reagent type (species) or resource | Designation | Source or reference | Identifiers | Additional information |
|---|---|---|---|---|
| Recombinant DNA reagent | Plasmid pSN672: pET21a(+) human KIF1A(1–393)-LZ-6His | *Anazawa et al., 2022* | RRID:Addgene_177362 | Template for SNAP-tag insertion |
| Recombinant DNA reagent | Plasmid pSN643: pET28a(+) human KIF1A(1–393)-LZ-mScarlet-I-Strep-tagII | *Anazawa et al., 2022* | — | Template for SNAP-tag insertion |
| Recombinant DNA reagent | Plasmid pSNAP-tag (T7)-2 vector | New England Biolabs | #N9181S | Template for SNAP-tag cDNA amplification |
| Recombinant DNA reagent | Plasmid pET21a(+) human KIF1A(1–393)-LZ-SNAP-6His | This study | — | Motor domain construct for force measurement |
| Recombinant DNA reagent | Plasmid pET28a(+) human KIF1A(1–393)-LZ-SNAP-Strep-tagII | This study | — | Used for heterodimer preparation |
| Recombinant DNA reagent | Plasmid pET21a(+) human KIF1A(1–393, P305L)-LZ-SNAP-6His | This study | — | KIF1A mutant construct associated with KAND individuals |
| Recombinant DNA reagent | Plasmid pET21a(+) human KIF1A(1–393, V8M)-LZ-SNAP-6His | This study | — | KIF1A mutant construct associated with KAND individuals |
| Recombinant DNA reagent | Plasmid pET21a(+) human KIF1A(1–393, A255V)-LZ-SNAP-6His | This study | — | KIF1A mutant construct associated with KAND individuals |
| Recombinant DNA reagent | Plasmid pYS05: KIF5B(1–560,G234A)::RA::SNAP::6His | *Shimamoto et al., 2015* | — | Inactive kinesin-5B mutant; gift from Dr. Yuta Shimamoto |
| Strain, strain background (*Escherichia coli*) | BL21(DE3) | Novagen | #69450 | |
| Peptide, recombinant protein | Tubulin (porcine brain) | Tokyo Shibaura Organ | — | Purified in-house |
| Peptide, recombinant protein | NeutrAvidin | Thermo Scientific | 31000 | DNA calibration rod immobilization |

*Continued on next page*

*Continued*

| Reagent type (species) or resource | Designation | Source or reference | Identifiers | Additional information |
|---|---|---|---|---|
| Peptide, recombinant protein | Biotinylated BSA | Thermo Scientific | 29130 | Surface coating |
| Chemical compound, drug | IPTG | Sigma-Aldrich | — | Protein expression inducer |
| Chemical compound, drug | Taxol | FUJIFILM Wako | 163–28163 | Microtubule stabilization |
| Chemical compound, drug | Casein | Sigma-Aldrich | C5890-500G | Surface blocking |
| Chemical compound, drug | BG-GLA-NHS | New England Biolabs | S9151S | SNAP-tag labeling |
| Chemical compound, drug | d-Desthiobiotin | Sigma-Aldrich | D1411 | Strep-tag elution |
| Chemical compound, drug | K10–PEG5K | Alamanda Polymers | mPEG5K-b-PLKC10 | DNA origami surface passivation |
| Sequence-based reagent | p8064 (DNA scaffold) | Tilibit Nanosystems | — | DNA origami scaffold |
| Sequence-based reagent | Core and handle staples (oligonucleotides) | Integrated DNA Technologies | — | Sequences in Tables S2–S7 |
| Sequence-based reagent | DNA nanospring (DNA nanostructure) | This study | — | Designed with caDNAno |
| Sequence-based reagent | DNA calibration rod (DNA nanostructure) | *Matsubara et al., 2023* | — | Used for force calibration |
| Software, algorithm | caDNAno | *Douglas et al., 2009* | — | DNA origami design |
| Software, algorithm | High Speed Recording | Hamamatsu Photonics | — | Video acquisition |
| Software, algorithm | Python | Python Software Foundation | RRID:SCR_008394 | Version 3.11.3 |
| Software, algorithm | Chain fitting analysis | This study | — | Custom Python code using scipy.optimize |

## Plasmid constructs

The plasmids used in this study are listed in *Supplementary file 1*. A recombinant human KIF1A(1–393)-LZ-SNAP-6His construct containing the motor domain (amino acids 1–361), neck linker, neck coil domain, and a GCN4 leucine zipper (LZ) for stabilized dimerization was used for NS force generated study (*Figure 1a*). To generate this construct, a human KIF1A(1–393)-LZ-6His construct named pSN672 (Addgene #177362) (*Anazawa et al., 2022*) was modified as follows: SNAP-tag cDNA was amplified by polymerase chain reaction (PCR) with high-fidelity PCR enzyme (Prime STAR MAX, Takara Bio Inc, Kusatsu, Japan, Cat#R045), using pSNAP-tag(T7)-2 vector (New England Biolabs, N9181S) as a template. Then the PCR product added with XhoI sites at both ends was inserted into the XhoI site of the plasmid pSN672 by using In-Fusion HD Cloning Kit (Takara Bio Inc, Cat#639649). This KIF1A (1–393)-LZ-SNAP-6His construct was then used as a template to generate the three aberrant motors by site-directed mutagenesis using the KOD Plus mutagenesis kit (TOYOBO, Cat#SMK-101). A recombinant human KIF1A(1–393)-LZ-SNAP-Strep-tagII construct was used to develop heterodimers of wild-type and mutant KIF1A. To create this KIF1A motor domain construct, a human KIF1A(1–393)-LZ-mScarlet-I-Strep-tagII construct, namely pSN643, was modified as follows: The mScarlet-I sequence of pSN643 (*Anazawa et al., 2022*) was replaced by SNAP cDNA sequence. All constructs were confirmed by sequencing. Inactive mutant kinesin 5 B (G234A) expression vector was a gift from Dr. Yuta Shimamoto (National Institute of Genetics), which has been described (pYS05 K560 pET-17b KIF5B (1–560, G234A)::LZ::His-tag) (*Rice et al., 1999*; *Shimamoto et al., 2015*).

## Recombinant protein expression and purification (homodimers)

Proteins were expressed in BL21(DE3) and SNAP-tagged KIF1A motor domain homodimers and inactive mutant KIF5B (G234A) homodimer were purified as previously described (*Anazawa et al., 2022*). Briefly, His-tagged SNAP-tagged KIF1A motor domain plasmids were transformed into BL21(DE3) (Novagen #69450), and the cells were cultured on LB agar supplemented with ampicillin at 37°C overnight. Colonies were picked and cultured in 10 ml LB medium supplemented with ampicillin overnight. Next morning, the culture was transferred to 750 ml of 2.5× YT (20 g/l Tryptone, 12.5 g/l

Yeast Extract, 6.5 g/l NaCl) supplemented with 10 mM phosphate buffer (pH 7.4) and 100 µg/ml ampicillin in 2 l flasks and shaken at 37°C. Two flasks were routinely prepared. When $OD_{600}$ reached 0.6, the flasks were cooled in ice-cold water for 30 min, then isopropyl-β-D-thiogalactoside was added to the cooled culture to final concentration of 0.2 mM to induce expression. The culture was shaken at 18°C overnight (~16 hr), and the bacteria expressing recombinant proteins were harvested by centrifugation at 4000 rpm, 20 min, 4°C (HITACHI himac CR-21). The cell pellet was resuspended in phosphate-buffered saline; PBS(−) and centrifuged again (4000 rpm, 20 min, 4°C). Supernatant was discarded completely, then cell pellets were frozen and stored at –80°C until use. To purify recombinant protein, bacteria pellets were resuspended in protein buffer (50 mM HEPES, pH 8.0, 150 mM $KCH_3COO$, 2 mM $MgSO_4$, 10% glycerol), supplemented with 1 mM ATP, protease inhibitors, such as 4-(2-Aminoethyl) benzenesulfonylfluoride (FUJIFILM), Leupeptin (FUJIFILM, Wako #336-40413), Aprotinin from Bovine Lung (FUJIFILM, Wako #013-28311), and Pepstatin A (FUJIFILM, Wako #330-43973), using 5 ml protein buffer per gram of wet cell paste. After gentle vortex, bacteria resuspension was with Benzonase nuclease (Novagen) and Lysozyme (Merch), followed by sonication using Branson SONIFIER 250. Lysate was obtained by centrifugation (40,000 rpm, 30 min, 4°C, HITACHI himac CP-80α). After centrifugation, the supernatant was loaded on TALON Metal Affinity Resin (Takara Bio Inc, Cat#635502), and incubated for 30 min at 4°C with mild shaking under platform shaker. The resin was washed twice with His-tag wash buffer (50 mM HEPES, pH 8.0, 450 mM $KCH_3COO$, 2 mM $MgSO_4$, 10% glycerol, 10 mM imidazole), containing 100 µM ATP. Protein was eluted with 10 ml His-tag elution buffer (50 mM HEPES, pH 8.0, 150 mM $KCH_3COO$, 2 mM $MgSO_4$, 10% glycerol, 500 mM imidazole) containing 100 µM ATP. Eluted fractions were collected and concentrated to ~300 µl using Amicon Ultra centrifugal filters (Merck, Darmstadt, Germany). The affinity-purified protein was further separated by gel filtration using a Superdex 200 Increase 10/300 GL column (Cytiva) in protein buffer on an NGC Chromatography system (Bio-Rad Laboratories). Peak fractions were collected and aliquoted and snap-frozen in liquid nitrogen. Protein concentration was assessed by Bradford method (Bio-Rad Protein assay dye Reagent Concentrates, Cat#5000006). Each step of purification was analyzed by SDS–polyacrylamide gel electrophoresis (SDS–PAGE) (*Figure 1—figure supplement 1*). All recombinant DNA experiments conducted in accordance with the Cartagena Protocol on Biosafety and all procedures were approved by the Genetic Modification Safety Committee of Yamaguchi University (J18016). Preparation of all plasmids and recombinant proteins was performed at Yamaguchi University Center for Gene Research.

## Purification of heterodimers

BL21(DE3) cells transformed with KIF1A(1–393)::LZ::SNAP:: Strep-tag II plasmid were cultured in LB supplemented with kanamycin at 37°C. Competent cells were prepared using a Mix&Go kit (Zymogen). The competent cells were further transformed with mutant KIF1A(1–393)::LZ::SNAP::His plasmid and selected on LB agar supplemented with ampicillin and kanamycin. Colonies were picked and cultured in 10 ml LB medium supplemented with ampicillin and kanamycin overnight. Next morning, 10 ml of the medium was transferred to 750 ml 2.5× YT supplemented with carbenicillin and kanamycin in a 2-l flask and shaken at 37°C. Two flasks were routinely prepared. The procedures for protein expression in bacteria and preparation of bacterial lysate were the same as for the purification of homodimers except that the component of purification buffer was 50 mM HEPES, pH 8.0, 150 mM $KCH_3COO$, 2 mM $MgSO_4$, 1 mM EGTA, 10% glycerol. Lysate was loaded on Strep-Tactin resin (IBA, GmbH)(bead volume: 2 ml) and incubated for 1 hr at 4°C with mild shaking under platform shaker. The resin was washed with 40 ml wash buffer. Protein was eluted with 10 ml protein buffer supplemented with 2.5 mM d-Desthibiotin (Merck, Sigma-Aldrich, D1411). The eluted solution was then loaded on Ni-NTA agarose (QIAGEN, #30210) (bead volume: 2 ml) and incubated for 1 hr at 4°C with mild shaking under platform shaker. The resin was washed with 40 ml His-tag wash buffer (50 mM HEPES, pH 8.0, 450 mM $KCH_3COO$, 2 mM $MgSO_4$, 10 mM imidazole, 10% glycerol), containing 100 µM ATP. Protein was eluted with His-tag elution buffer (50 mM HEPES, pH 8.0, 150 mM $KCH_3COO$, 2 mM $MgSO_4$, 10% glycerol, 500 mM imidazole) containing 100 µM ATP. Eluted fractions were collected and concentrated to ~300 µl using Amicon Ultra centrifugal filters (Merck), then subjected to separation on NGC chromatography system (Bio-Rad) equipped with a Superdex 200 Increase 10/300 GL column (Cytiva). Peak fractions were pooled, concentrated in an Amicon filter again, and snap frozen in liquid nitrogen. The concentration and

quality of the protein were assessed by Bradford method and SDS–PAGE, respectively (*Figure 1—figure supplement 2*).

## Microtubule preparation

Tubulin was purified from porcine brain (Tokyo Shibaura Organ Co, Ltd) through two cycles of polymerization and depolymerization using the revised method of reference (*Castoldi and Popov, 2003*), and stored at −80°C in PEM buffer (100 mM PIPES, 1 mM MgCl$_2$, 1 mM EGTA, pH 6.9, adjusted with KOH). For polymerization, the tubulin solution was mixed with 2 mM GTP, 10 mM MgCl$_2$, and 15% DMSO, and was then incubated at 37°C for 30 min. Prior to polymerization, the tubulin solution was centrifuged at 180,000 × *g* for 5 min at 2°C, and the supernatant was collected. The polymerized solution was then incubated at room temperature for 20 min in BRB80 buffer (80 mM PIPES, 1 mM MgCl$_2$, 1 mM EGTA, pH 6.8 w/ KOH) supplemented with 10 µM Taxol (163-28163; FUJIFILM) to stabilize the microtubules. The microtubules were pelleted by centrifugation and then resuspended in BRB80 buffer. Microtubules were stored in the dark at room temperature for up to 2 weeks.

## Preparation of NSs and DNA calibration rod

NSs were designed using caDNAno software (*Douglas et al., 2009*), as described in the method from the reference (*Matsubara et al., 2023*). Briefly, NSs were designed by introducing a negative superhelical strain into a four-helix bundle arranged on a square lattice. To impose the torsional strain, two additional nucleotides were periodically inserted every 32 bp in the upper pair of helices, while two nucleotides were deleted at the corresponding positions in the lower pair of helices. This design yielded a uniform coil structure with a diameter of ~35 nm. For the kinesin–NS conjugation, 32-base single-stranded DNA (ssDNA) handles (handle 32A and 32B in *Supplementary file 2*) were extended from both ends of the NS. To construct the NS, 10 nM scaffold (p8064, tilibit nanosystems) was mixed with 100 nM core staples (*Supplementary file 3*), 100 nM handle staples (*Supplementary file 2*), and 12.4 mM Cy3-labeled antihandle (*Supplementary file 4*). The folding reaction was carried out in folding buffer (5 mM Tris pH 8.0, 1 mM EDTA, and 14 mM MgCl$_2$) with rapid heating to 80°C and cooling in single degree steps to 60°C over 2 hr followed by additional cooling in single degree steps to 25°C over another 24 hr. DNA calibration rods were also prepared as previously described (*Matsubara et al., 2023*). Briefly, 10 nM scaffold (p8064) was mixed with 100 nM core staples (*Supplementary file 5*), 100 nM handle staples (*Supplementary file 6*), 12.4 mM Cy3-labeled antihandle and 800 nM biotin-labeled antihandle (*Supplementary file 7*). The folding reaction was carried out in folding buffer (5 mM Tris pH 8.0, 1 mM EDTA, and 14 mM MgCl$_2$), and the heating and cooling condition was the same as that for the NS. The folded DNA nanostructures were purified by glycerol gradient ultracentrifugation as previously described (*Matsubara et al., 2023*). Briefly, 15–45% (vol/vol) gradient glycerol solutions in buffer A (1× TE buffer containing 11 mM MgCl$_2$) were made, and the glycerol fractions containing monomeric nanostructures were determined by agarose gel electrophoresis. The concentration of the nanostructures was determined with a Nanodrop spectrophotometer (Thermo Scientific), and the solution was aliquoted and stored at −80°C until use. To minimize electrostatic interference between NSs and microtubules, NSs were coated with oligolysine conjugated to PEG (K10 PEG5K, purchased from Alamanda Polymers) (*Castoldi and Popov, 2003*) as follows. 10 µl of 4.2 nM NSs were mixed with 0.45 µl of K10–PEG5K at a concentration corresponding to an N:P ratio (ratio of nitrogen in amines to phosphates in DNA) of 0.5:1. The mixture was then incubated at room temperature for 30 min.

## Kinesin–NS conjugation

A 32-base amine-modified DNA oligonucleotide (Hokkaido System Science) (oligo 32A* and 32B* in *Supplementary file 2*) was conjugated with BG-GLA-NHS (S9151S; New England Biolabs) through the reaction between the ester and amine groups (BG-oligonucleotide). Here, the oligo 32A* and 32B* was complementary to the ssDNA handle sequence (handle 32A and 32B) of the NS. Then, the 35 µM BG-oligonucleotide (2 µl) and 1 µM SNAP-tagged KIF1A (50 µl) or 1 µM SNAP-tagged inert KIF5B mutant (G234A) (50 µl) were mixed and incubated at room temperature for 1 hr. The oligo-labeled kinesin was aliquoted and stored at −80°C. When the oligo-labeled kinesin and NS were conjugated, 5 µl of oligo 32A*-labeled KIF1A or oligo 32B*-labeled inert KIF5B (0.5–1 µM) was mixed

with 10 µl of NSs (~5 nM) in SRP90 buffer (90 mM HEPES, 50 mM CH$_3$COOK, 2 mM Mg(CH$_3$COO)$_2$, 1 mM EGTA, pH 7.6, adjusted with KOH), and incubated on ice for 30 min.

## Single-molecule experiment on kinesin–NS conjugates

A flow chamber was constructed from two uncoated coverslips of different sizes: an 18 mm × 18 mm coverslip (2918COVER18-18, Iwaki) was placed on top, and a 24 mm × 24 mm coverslip (C024241, Matsunami) was placed at the bottom, separated by two spacers of ~50 µm thickness, resulting in a volume of approximately 10 µl. Unless otherwise noted, the assay buffer was the SRP90 buffer supplemented with 20 µM Taxol, 0.4 mM 2-mercaptoethanol (131-14572, FUJIFILM), and 0.2 mg/ml casein (C5890-500G, Sigma-Aldrich). Microtubules were diluted in the buffer (without casein) and flowed into the chamber, allowing them to attach to the glass surface of the chamber through nonspecific interaction during a 5-min incubation. Then, the chamber was incubated with the blocking buffer (assay buffer containing 0.5 mg/ml casein) for 5 min to prevent nonspecific binding, during which microtubules that did not adhere to the glass surface were washed away using the blocking buffer. Next, the assay buffer containing kinesin–NS conjugates was flowed into the chamber, followed by a 5-min incubation to allow the kinesin–NS conjugates to attach to the microtubules. Finally, the assay buffer, additionally supplemented with 1 mM ATP, 3 mg/ml glucose, 0.1 mg/ml glucose oxidase, and 0.02 mg/ml catalase, was flowed into the chamber. Note that the latter three regents are an oxygen scavenger system for fluorescence antifade protection. This step allowed for the removal of kinesin–NS conjugates that had not attached to the microtubules. Finally, to prevent the evaporation of the buffer inside the chamber during the experiment, the open ends of the flow chamber were sealed with clear nail polish. The motion of kinesin–NS conjugates was observed using a fluorescence microscope (IX83, EVIDENT) equipped with a 100×oil immersion objective lens (UPlanFL N 100×/1.30 Oil, EVIDENT) and a fluorescence mirror unit (TRITC-B, Semrock), and was recorded at 33 frames per second (fps) with an effective pixel size of 65 nm by a CMOS camera (ORCA-Flash4.0, Hamamatsu Photonics). Experiments were also performed at 100 fps in addition to the recording rate of 33 fps (*Figure 3—figure supplement 1*). The flow chamber was maintained at 25°C using a heating plate (HP-R-Z002, Live Cell Instrument) during observation.

## Observation of DNA calibration rod

A flow chamber was prepared in the same manner as described for the observation of kinesin–NS conjugates. Biotinylated BSA (0.5 mg/ml, 29130, Thermo Scientific) was incubated for 5 min to coat the glass surface of the chamber. Excess biotinylated BSA was washed by the assay buffer (SRP90 buffer supplemented with 10 mM Mg(CH$_3$COO)$_2$). Subsequently, neutravidin (0.5 mg/ml, 31000, Thermo Scientific) was flowed into the chamber and incubated for 3 min. Unbound neutravidin was removed by the assay buffer. Finally, ~6 pM biotinylated DNA calibration rods (*Matsubara et al., 2023*) were flowed into the chamber and incubated for 5 min to allow their attachment to the glass surface. Unbound DNA calibration rods were washed out by the assay buffer containing the oxygen scavenger system for fluorescence antifade protection (3 mg/ml glucose, 0.1 mg/ml glucose oxidase, and 0.02 mg/ml catalase).

## Conversion from image to force

Video acquisition was conducted using the High Speed Recording software (Hamamatsu Photonics), and the exported 8-bit video data in uncompressed AVI format were imported into a custom-written Python program for further analysis. The fluorescent spots corresponding to the DNA calibration rods and NSs were analyzed using the chain fit method (*Equation 2*). To estimate their lengths ($L$) using *Equation 2*, parameter fitting was performed using curve fitting via the 'curve_fit' function from the 'scipy.optimize' module of Python (version 3.11.3). Data acquisition of the NS fluorescence images was performed under the same optical conditions (e.g., excitation laser intensity, dichroic mirror, filter, and camera settings) as those used for the DNA calibration rod. A correction was applied to the NS's length ($L$) to account for possible bending of the DNA origami structure due to thermal fluctuations (*Figure 2e, f*). The modified length $L_m^C$ is calculated as $L_m^C = 1.07 \times L - 2.35$ where $L$ is the estimated value by the chain fitting method (*Equation 2*). For the parameter $\sigma_{\text{long}}$ estimated by the Gaussian fitting method (*Equation 1*), the length $L_m^G$ is calculated as $L_m^G = 3.67 \times \sigma_{\text{long}} - 224$. These empirical linear relationships between $L_m^C$ and $L$, and between $L_m^G$ and $L$ were obtained from the DNA

calibration rod experiment presented in *Figure 2f*. Then, the force value corresponding to the length value ($L_m^C$ or $L_m^G$) was determined from the force–extension curve (*Figure 1d* of the main text). Here, the force–extension curve was fitted with an exponential function ($F(x) = 4.08 \times 10^{-4} \cdot \exp(16.6 \cdot x)$), where $x$ is the extension in μm and $F(x)$ is the force in pN.

### Angular fluctuation

Let $\theta_t$ denotes the angle between the major axes of a microtubule and an NS images (the major axis of the ellipse fitted to the fluorescence image of the NS) in frame $t$. Then angle $\theta_t$ is converted to $\phi_t = 2\theta_t \in [-\pi, \pi]$, and mapped to a unit vector $\overrightarrow{v_t} = (\cos\phi_t, \sin\phi_t)$. The mean of $\overrightarrow{v_t}$ over 10 frames is defined as $\overrightarrow{R_t} = \frac{1}{\omega} \sum_{j=t-\omega/2}^{t+\omega/2+1} \overrightarrow{v_t}$. The angular fluctuation is quantified as $1 - \left|\overrightarrow{R_t}\right|$. Here, the relation $1 - \left|\overrightarrow{R_t}\right| \sim 0$ indicates that an NS is aligned with the direction of a microtubule, while $1 - \left|\overrightarrow{R_t}\right|$ approaching 1 indicates that the NS is oriented away from the direction of the microtubule. We considered the NS to be attached to the microtubule when $1 - \left|\overrightarrow{R_t}\right| < 0.1$. If this condition lasted for at least 5 consecutive frames, the corresponding time interval was measured as the attachment duration.

### Rate of relative increase in NS's length

The attachment of KIF1A to the microtubule was identified by the marked suppression of angular fluctuations of the NS relative to the microtubule, which clearly distinguished the bound state from the diffusive search behavior (e.g., red regions in *Figure 3a*). Within this attachment duration, the force plateau was detected by analyzing the temporal change in NS length (e.g., violet regions in *Figure 3a*): the 10-frame moving average of the extension, $\bar{L}(\tau) \left(= \left(\sum_{t=10(\tau-1)+1}^{10\tau} L(t)\right)/10\right)$ where $t$ represents a video frame, was computed, and the regions where the relative increase rate $(\bar{L}(\tau) - \bar{L}(\tau-1))/\bar{L}(\tau-1) \leq 0.02$ were regarded as plateau segments. The mean NS extension over these plateau segments was defined as $L_{\text{stall}}$.

### Statistical analysis

Statistical analyses were performed using Python (version 3.11.3) with scipy.stats (for Shapiro–Wilk, Levene, Mann–Whitney $U$, and Kruskal–Wallis tests) and scikit-posthocs (for Dunn's post hoc test with Bonferroni correction). Statistical comparisons were conducted for $L_{\text{stall}}$ and $\Delta t$ measurements. The normality of each dataset was assessed using the Shapiro–Wilk test. All datasets were non-normally distributed, except for $L_{\text{stall}}$ values obtained from the homozygous P305L mutant (P305L/P305L) and heterozygous V8M mutant (V8M/WT). Homogeneity of variances was evaluated using Levene's test. Homogeneity of variance was confirmed for $L_{\text{stall}}$ between the w/o PEG and w/ PEG conditions, and among WT/WT and A255V genotypes. However, variance differed significantly among WT/WT and P305L genotypes and among WT/WT and V8M genotypes. All $\Delta t$ datasets showed unequal variances. Based on these results, non-parametric statistical tests were applied. The Mann–Whitney $U$ test was used for comparing $L_{\text{stall}}$ between w/o PEG and w/ PEG, and for comparing $\Delta t$ between homozygous and heterozygous mutant combinations. The Kruskal–Wallis test was applied to compare $L_{\text{stall}}$ across WT/WT, mutant/mutant, and mutant/WT combinations for each KIF1A mutant (P305L, V8M, and A255V), followed by Dunn's test with Bonferroni correction.

## Acknowledgements

We acknowledge Dr. Yuta Shimamoto for providing plasmids, and Ms. Kimiko Nagino and Mr. Gai Ohashi for purifying microtubules. This work was supported by JSPS KAKENHI (Grant No. 23H02442), the Precise Measurement Technology Promotion Foundation (PMTP-F), and a KIF1A.org Mini Grant to KH, JSPS KAKENHI (Grant No. 21H01053) and JST, CREST (Grant No. JPMJCR2023) to MI, JST PRESTO (Grant No. JPMJPR21E2) to TA.

## Additional information

### Funding

| Funder | Grant reference number | Author |
|---|---|---|
| Japan Society for the Promotion of Science | 23H02442 | Kumiko Hayashi |
| Japan Society for the Promotion of Science | 21H01053 | Mitsuhiro Iwaki |
| Japan Science and Technology Agency | 10.52926/jpmjcr2023 | Mitsuhiro Iwaki |
| Japan Science and Technology Agency | 10.52926/JPMJPR21E2 | Takayuki Ariga |

The funders had no role in study design, data collection, and interpretation, or the decision to submit the work for publication.

### Author contributions

Nobumichi Takamatsu, Data curation, Software, Formal analysis, Methodology, Writing – original draft; Hiroko Furumoto, Resources, Writing – original draft, Writing – review and editing; Takayuki Ariga, Conceptualization, Resources, Funding acquisition, Writing – review and editing; Mitsuhiro Iwaki, Conceptualization, Resources, Funding acquisition, Writing – original draft, Project administration, Writing – review and editing; Kumiko Hayashi, Conceptualization, Supervision, Funding acquisition, Writing – original draft, Project administration

### Author ORCIDs

Nobumichi Takamatsu 
Hiroko Furumoto 
Takayuki Ariga 
Mitsuhiro Iwaki 
Kumiko Hayashi 

Reviewer #1 (Public review): https://doi.org/10.7554/eLife.108477.3.sa1
Reviewer #2 (Public review): https://doi.org/10.7554/eLife.108477.3.sa2
Author response https://doi.org/10.7554/eLife.108477.3.sa3

## Additional files

### Supplementary files

Supplementary file 1. Plasmid list. Listed below are the plasmids used in this study.

Supplementary file 2. Handle staples and anti-handles used to link KIF1A and inert KIF5B at the end of NSs. In the sequences of staples handle 32A and 32B, italicized regions indicate single-stranded DNA (ssDNA) handle sequences.

Supplementary file 3. Core staples and handle staples for Cy3 to build the NS. Sequences in italics indicate the handle site.

Supplementary file 4. Antihandle carrying Cy3 to label the NS.

Supplementary file 5. Core staples to build the DNA calibration rod.

Supplementary file 6. Handle staples for the DNA calibration rod. Sequences in italics indicate the handle site.

Supplementary file 7. Antihandles for DNA calibration rod.

MDAR checklist

## Data availability

All data generated or analyzed during this study are included in the manuscript and supporting files; source data files have been provided. The primary dataset has been deposited in Dryad (https://doi.org/10.5061/dryad.6q573n6d4).

The following dataset was generated:

| Author(s) | Year | Dataset title | Dataset URL | Database and Identifier |
|---|---|---|---|---|
| Takamatsu N, Furumoto H, Ariga T, Iwaki M, Hayashi K | 2026 | Data from: Stall force measurement of the kinesin-3 motor KIF1A using a programmable DNA origami nanospring | https://doi.org/10.5061/dryad.6q573n6d4 | Dryad Digital Repository, 10.5061/dryad.6q573n6d4 |

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
