## [Editor Report · eLife Assessment]

Optical tweezers have been instrumental to the determination of mechanical parameters of molecular motors. This study by Takamatsu et al. reports key mechanical parameters of kinesin KIF1A using fluorescence microscopy, wherein the motor is tethered to a DNA nanospring, without the use of an optical trapping apparatus, which represents an exciting development. The approach and the findings reported change current thinking about KIF1A‑mediated transport, with potential implications for understanding human disease. The findings are **important** and the strength of the evidence is **compelling**.

---

## [Referee Report · Reviewer #1 (Public review)]

Summary:

This study uses a novel DNA origami nanospring to measure the stall force and other mechanical parameters of the kinesin-3 family member, KIF1A, using light microscopy. The key is to use SNAP tags to tether a defined nanospring between a motor-dead mutant of KIF5B and the KIF1A to be integrated. The mutant KIF5B binds tightly to a subunit of the microtubule without stepping, thus creating resistance to the processive advancement of the active KIF1A. The nanospring is conjugated with 124 Cy3 dyes, which allows it to be imaged by fluorescence microscopy. Acoustic force spectroscopy was used to measure the relationship between the extension of the NS and force as a calibration. Two different fitting methods are described to measure the length of the extension of the NS from its initial diffraction-limited spot. By measuring the extension of the NS during an experiment, the authors can determine the stall force. The attachment duration of the active motor is measured from the suppression of lateral movement that occurs when the KIF1A is attached and moving. There are numerous advantages of this technology for the study of single molecules of kinesin over previous studies using optical tweezers. First, it can be done using simple fluorescence microscopy and does not require the level of sophistication and expense needed to construct an optical tweezer apparatus. Second, the force that is experienced by the moving KIF1A is parallel to the plane of the microtubule. This regime can be achieved using a dual beam optical tweezer set-up, but in the more commonly used single-beam set-up, much of the force experienced by the kinesin is perpendicular to the microtubule. Recent studies have shown markedly different mechanical behaviors of kinesin when interrogated by the two different optical tweezer configurations. The data in the current manuscript are consistent with those obtained using the dual-beam optical tweezer set-up. In addition, the authors study the mechanical behavior of several mutants of KIF1A that are associated with KIF1A-associated neurological disorder (KAND).

Strengths:

The technique should be cheaper and less technically challenging than optical tweezer microscopy to measure the mechanical parameters of molecular motors. The method is described in sufficient detail to allow its use in other labs. It should have a higher throughput than other methods.

Weaknesses:

The experimenter does not get a "real-time" view of the data as it is collected, which you get from the screen of an optical tweezer set-up. Rather, you have to put the data through the fitting routines to determine the length of the nanospring in order to generate the graphs of extension (force) vs time. No attempts were made to analyze the periods where the motor is actually moving to determine step-size or force-velocity relationships.

Comments on revisions:

I am satisfied with the revision made by the authors in response to my first round of criticisms.

---

## [Referee Report · Reviewer #2 (Public review)]

Summary:

This work is important in my view because it complements other single-molecule mechanics approaches, in particular optical trapping, which inevitably exerts off-axis loads. The nanospring method has its own weaknesses (individual steps cannot be seen), but it brings new clarity to our picture of KIF1A and will influence future thinking on the kinesins-3 and on kinesins in general.

Strengths:

By tethering single copies of the kinesin-3 dimer under test via a DNA nanospring to a strong binding mutant dimer of kinesin-1, the forces developed and experienced by the motor are constrained into a single axis, parallel to the microtubule axis. The method is imaging-based which should improve accessibility. In principle, at least, several single-motor molecules can be simultaneously tested. The arrangement ensures that only single molecules can contribute. Controls establish that the DNA nanospring is not itself interacting appreciably with the microtubule. Forces are convincingly calibrated and reading the length of the nanospring by fitting to the oblate fluorescent spot is carefully validated. The excursions of the wild type KIF1A leucine zipper-stabilised dimer are compared with those of neuropathic KIF1A mutants. These mutants can walk to a stall plateau, but the force is much reduced. The forces from mutant/WT heterodimers are also reduced.

Weaknesses:

The tethered nanospring method has some weaknesses; it only allows the stall force to be measured in the case that a stall plateau is achieved, and the thermal noise means that individual steps are not apparent. The nanospring does not behave like a Hookean spring - instead linearly increasing force is reported by exponentially smaller extensions of the nanospring under tension. The estimated stall force for Kif1A (3.8 pN) is in line with measurements made using 3 bead optical trapping, but those earlier measurements were not of a stall plateau, but rather of limiting termination (detachment) force, without a stall plateau.

Comments on revisions:

The authors have successfully addressed my previous criticisms.

---

## [Author Response]

The following is the authors’ response to the original reviews.

**Public Reviews:**

**Reviewer #1 (Public review):**

We thank Reviewer #1 for the careful reading of our manuscript and for the constructive comments. We have provided responses to each of the comments below.

We greatly appreciate Reviewer #1’s accurate public review of our study on the kinesin motor using the DNA origami nanospring (NS). With respect to the strengths, we fully agree with Reviewer #1’s comments. Regarding the weakness, we would like to respond as follows.

It is true that, unlike optical tweezers, our method does not provide real-time data display. Optical tweezers enable real-time observation and manipulation of kinesin molecules at arbitrary time points. Achieving real-time observation and manipulation is indeed an important challenge for the future development of the NS technique. On the other hand, Iwaki et al. (our co-corresponding author) has already investigated dynamic properties of motor proteins under load, such as step size and force–velocity relationship of myosin VI using NS. We are now preparing high spatiotemporal resolution microscopy experiments on the KIF1A system to measure its step size and force–velocity relationship, which inherently require such resolution.

**Reviewer #2 Public Review**

We appreciate the constructive comments of Reviewer #2, which have strengthened both the presentation and interpretation of our results.

We would like to thank Reviewer #2 for providing a highly accurate assessment of the strengths of our experiments. Regarding the weaknesses, we would like to respond as follows. First, Iwaki et al. (our co-corresponding author) have already succeeded in observing the stepping motion of myosin VI using the nanospring (NS) in their previous work. We are also currently preparing high spatiotemporal resolution microscopy experiments to observe the stepping motion of KIF1A in our system. Second, while it is true that the NS does not follow Hooke’s law, it is possible to design and construct NSs with an appropriate dynamic range by tuning the spring constant to match the forces exerted by protein molecules. Finally, we agree that our first observation of the stall plateau in KIF1A using the NS is a meaningful achievement. However, with respect to the suggestion that “increasing validity requires also studying kinesin-1,” we have a somewhat different perspective. The validity of the NS method has already been thoroughly examined in the previous work on myosin VI by Iwaki et al., where results were compared with those obtained using optical tweezers. Moreover, the focus of this manuscript is on KAND caused by KIF1A mutations. From this perspective, although we appreciate the suggestion, we consider it important to keep the present study focused on KIF1A and its implications for KAND.

**Recommendations for the authors:**

**Reviewer #1 (Recommendations for the authors):**
(1) The authors detect the attachments that occur during a processive run by KIF1A by monitoring the suppression of the angular fluctuations of the fluorescent signal and plot this, for example, in Figure 3a as the Length of the NS (which presumably is a readout of force) vs time. This interval includes the time when the KIF1A is actively moving along the MT and when it is stalled. It would be interesting to know the actual stall time of the motor in order to be able to calculate a detachment rate constant. For attachment periods such as the first example highlighted in pink in Figure 3a, the stall time is pretty much equal to the attachment time since the motor is moving so fast and the stall period is so long. However, for short attachment times such as the fifth pink interval shown in this same figure or the traces with the mutant KIF1As in Figure 4 this is not so. Can the authors institute a program to identify the periods where the motor has stretched the NS spring to the point where it stalls, and then calculate this time in order to do an exponential fit to the "dwell time distribution"?

By introducing another criterion (see Methods, “Rate of relative increase in NS’s length”), the attachment duration was separated into the two time regions noted by the reviewer. After reanalyzing all the data, we evaluated only the stall duration this time. As a result, the estimated stall-force values became more reliable and accurate. The dwell time analysis of was performed and included in the supplementary material for WT KIF1A, for which sufficient data were available.

(2) The histogram of stall events in Figure 3b is quite broad. Please discuss.

The newly added distributions from individual molecules (Fig. 3b) show that the variety in the stall force distribution is not due to multiple molecules, but is primarily an intrinsic property of single KIF1A molecules reflecting the complex kinetics of KIF1A under load, including occasional backward steps and reattachments. In addition, because the nanospring is a non-linear spring, a disadvantage is that even small fluctuations in extension can result in a substantial deviation in the measured stall force. These points have been added to the Discussion section.

(3) Figure 3c, it is clear that for attachment times greater than 5s the attachment duration is independent of the Lstall, but this is not so clear for the short attachment durations. Some of this may relate to the fact that you're measuring attachment durations and not stall or dwell times as described in my first comment. Do you feel this is due to less precision in measuring the "attachment duration" during the short attachments, or just simply that more data is needed here? I assume that you do not want to imply that there is a load-dependence of the attachment durations here? Perhaps an expanded view of the data set from 0-10 seconds would clarify.

As described in our response to comment (1), the stall durations were separated from the attachment durations. This improved the measurement accuracy and revealed that and are uncorrelated (Fig. 3c). We appreciate this constructive comment.

**Reviewer #2 (Recommendations for the authors):**
(1) Off-axis forces are described as 'upward', 'perpendicular', and 'horizontal'. Consider referring to off-axis force, and if necessary, defining the direction of the force(s) relative to the axis of the immobilised MT. If necessary, a cartoon of XYZ axes might be added to F1c?

An XZ axis was added to the schematic in Fig. 1c.

(2) If I understand correctly, stall forces are calculated by averaging the entire region in which the angular fluctuation is reduced below a threshold. In cases like the 3rd and 7th events on the trace in F1a, this will reduce the average. Perhaps consider separately averaging the later time points in each stall event? Perhaps also consider correlating the angular fluctuation signals and the spring length signal? Some fluctuations during stall plateaus might indicate slip back and re-engage events?

Instead of separately averaging the later time points in each stall event, we separated the stall force duration from the overall attachment duration (Fig. 3). This allowed us to obtain more accurate stall force values. The relationship between the NS length and the angular fluctuation during KIF1A slip-back events differed among individual stall events, and no clear trend was observed. Two representative examples are shown in the Author response image 1.

(3) Please describe all relevant methods fully instead of referencing previous work. For example, nanospring preparation refers readers to reference 21 (which in turn references an earlier paper).

We revised the Methods section to include the procedures described in the previous reference, and we added the sequence information of the DNA origami to the supplementary information.

(4) Were any experiments tried at reduced ATP concentration?(5) Were any data obtained from WT KIF5B? For kinesin-1, stall plateau forces of >7 pN are obtained.

This study focused on comparing the stall forces of wild-type and KAND-related mutant KIF1A molecules under physiological ATP conditions, as our main goal was to characterize the disease-relevant phenotypes. Experiments at reduced ATP concentrations and with WT KIF5B are indeed important future directions but are beyond the scope of the present study. These follow-up experiments are currently in progress.

(6) In Figure 1b, consider showing the attachment to the mutant KIF5B, and reversing the orientation so it corresponds to Figure 1c.

KIF1A and KIF5B share the same binding method, so to indicate that the schematic in Fig. 1b represents both, we replaced ‘KIF1A’ with ‘Kinesin’.

(7) In Figure 3d, add force axis. In general, please re-check all force axes. In Supplement S3, the stall plateau labels appear well above their corresponding axis ticks. In Figure 4, several mutants appear to be stalling at well over 5 pN, yet Table 1 gives a much lower value. Presumably, this reflects averaging effects?

We added the force axis to Fig. 3d. Besides, we corrected Fig. S3 and Fig. 4 because there were errors in the conversion from length to force. As the reviewer pointed out, the apparent discrepancy between the force values in Fig. 4 and Table 1 arises mainly from averaging effects.